# Immunoendocrine Dysregulation during Gestational Diabetes Mellitus: The Central Role of the Placenta

**DOI:** 10.3390/ijms22158087

**Published:** 2021-07-28

**Authors:** Andrea Olmos-Ortiz, Pilar Flores-Espinosa, Lorenza Díaz, Pilar Velázquez, Carlos Ramírez-Isarraraz, Verónica Zaga-Clavellina

**Affiliations:** 1Departamento de Inmunobioquímica, Instituto Nacional de Perinatología Isidro Espinosa de los Reyes (INPer), Ciudad de México 11000, Mexico; nut.aolmos@gmail.com (A.O.-O.); m.pilar.flores.e@gmail.com (P.F.-E.); 2Departamento de Biología de la Reproducción, Instituto Nacional de Ciencias Médicas y Nutrición Salvador Zubirán, Ciudad de México 14080, Mexico; lorenzadiaz@gmail.com; 3Departamento de Ginecología y Obstetricia, Hospital Ángeles México, Ciudad de México 11800, Mexico; m.pilarvs@hotmail.com; 4Clínica de Urología Ginecológica, Instituto Nacional de Perinatología Isidro Espinosa de los Reyes (INPer), Ciudad de México 11000, Mexico; drcarlos.ri@gmail.com; 5Departamento de Fisiología y Desarrollo Celular, Instituto Nacional de Perinatología Isidro Espinosa de los Reyes (INPer), Ciudad de México 11000, Mexico

**Keywords:** inflammation, cytokines, adipokines, antimicrobial peptides, oxidative stress, metabolic stress, IGF-I, insulin, lactotroph hormones, angiogenesis

## Abstract

Gestational Diabetes Mellitus (GDM) is a transitory metabolic condition caused by dysregulation triggered by intolerance to carbohydrates, dysfunction of beta-pancreatic and endothelial cells, and insulin resistance during pregnancy. However, this disease includes not only changes related to metabolic distress but also placental immunoendocrine adaptations, resulting in harmful effects to the mother and fetus. In this review, we focus on the placenta as an immuno-endocrine organ that can recognize and respond to the hyperglycemic environment. It synthesizes diverse chemicals that play a role in inflammation, innate defense, endocrine response, oxidative stress, and angiogenesis, all associated with different perinatal outcomes.

## 1. Introduction

Gestational Diabetes Mellitus (GDM) is a transient condition characterized by carbohydrate intolerance, hyperglycemia, peripheral insulin resistance, insufficient insulin secretion or activity, endothelial dysfunction, and low-grade inflammation during pregnancy, frequently between 24 and 28 weeks of gestation [1]. Although it is a transient, GDM effects can last beyond the perinatal period and impact the health of mother and fetus in both short- and long-term [2,3,4].

In 2017, it was estimated that 21.3 million births (16.2%) worldwide were affected by hyperglycemia during pregnancy, with GDM contributing 86.4% of these cases [5,6]. Furthermore, an increase in the prevalence of GDM effects is expected due to the parallel increasing rate of pre-gestational obesity and excessive weight gain during pregnancy.

It is well accepted that a key event in the onset of GDM is the maternal peripheral insulin resistance. During normal pregnancy, there is a transient and physiological state of decreased insulin sensitivity, necessary to prioritize fetal glucose uptake. In response, β-cells proliferate and synthesize more insulin as a mechanism to counteract insulin resistance and favor euglycemia. However, before pregnancy, some women have a first pancreatic hit by GDM risk factors such as pre-gestational overweight, obesity, hypercaloric diet, personal or familiar antecedent of GDM, advanced maternal age, or presence of insulin resistance disorders such as polycystic ovarian syndrome [1,7]. With pregnancy, these women are exposed to a second pancreatic hit: the insulin resistance associated with early pregnancy. These two hits lead to GDM development because of inadequate compensatory changes in β-cell mass activity and proliferation due to a more pronounced insulin resistance condition, particularly during the second and third trimester of pregnancy [8,9]. This transient metabolic stress on the pancreas during pregnancy, partially explains why GDM is associated with a higher risk of post-partum development of Type 2 Diabetes Mellitus (T2DM) in the mother and fetus [10].

The American College of Obstetricians and Gynecologists (ACOG) recognizes two types of GDM. GDM Class 1 (A1GDM) patients respond to diet intervention (low glycemic index meals with low simple sugar and high fiber content) and exercise. GDM Class 2 (A2GDM) patients need pharmacologic treatment to achieve target glucose levels. For A2GDM, the first-line therapy recommended by the American Diabetes Association (ADA) is insulin, preferentially short-acting insulin (i.e., Lispro or Aspart), and long-acting insulin (i.e., Glargine or Detemir) [7,11]. However, other Societies including ACOG, German Diabetes Association, German Society of Gynecology and Obstetrics, and The Society for Maternal-Fetal Medicine recommend metformin instead [1,12]. Recent studies showed a lower risk for preeclampsia, macrosomia, neonatal hypoglycemia, and hypertensive disorders as well as better outcomes in maternal weight gain and glycemic control. No difference was observed in rates of caesarean section, neonatal respiratory distress and preterm birth compared to insulin treatment alone [13,14,15,16,17]. There is insufficient evidence on the long-term effects of prenatal exposure to metformin (especially because it crosses the placenta). Two recent studies showed no difference in growth and development in children of metformin-treated and insulin-treated mothers over a four-year period [18,19]. More long-term studies are needed to understand the long-term effects of metformin during pregnancy.

As it would be expected, hyperglycemia and GDM disturb placental ultrastructure and morphophysiology since the early stages of the disease. Reported placental abnormalities in GDM patients include increased placental weight, intimate glycogen deposits, increased number of syncytial knots, villous edema, and larger syncytial area and volume for favoring nutrient uptake [20,21,22,23]. Histopathologic analysis of GDM placentae indicates enhanced angiogenesis and high vasculogenesis rate evidenced by increased villous vascularity often associated with thickened immature villi capillaries and signs of placental hypoperfusion [24,25]. There has also been reported increased fibrinoid necrosis, chorangiosis, and ischemia [25,26]. Additionally, GDM syncytiotrophoblasts present an exaggerated mitochondrial dysfunction accompanied by a lower rate of glycolysis, oxidative phosphorylation, and ATP synthesis, which indicates a compromised metabolic supply and therefore placental overstress [27,28]. Lipid metabolism of the placenta is also distorted in GDM with evidence of larger lipid droplets, higher triglyceride accumulation, and fatty acid transporter expression [28,29]. This is all indicative of modification of the endocrine, immune, angiogenic, and antioxidant functions by placentae in GDM mothers. In this review, we aim to understand the placenta’s role as a vital organ that acts as the interface between maternal and fetal metabolisms impacted by a pathological condition as GDM. In this scenario, the placenta as an immuno-endocrine organ is required to recognize and respond to the hyperglycemic environment by synthesizing diverse cytokines, chemokines, adipokines, antimicrobial peptides, and hormones.

## 2. Role of Placenta in the Endocrine Milieu of GDM

The endocrine system is the earliest system developing during intrauterine life. From the stage of two-blastomeres, the embryo begins to secrete the beta-fraction of the human chorionic gonadotrophin (β-hCG), which has been suggested to be a product of mRNAs previously stored in oocytes [30,31]. Later, by 6 days post-fertilization, the trophectoderm establishes its endocrine phenotype through the de novo synthesis of β-hCG. The early placenta then turns on its hormonal switch and maintains secretion of a broad panel of hormones with central activities in the maintenance of pregnancy, fetal growth, and development. These include somatostatin, placental lactogens, placental growth hormone (PlGF), gonadotropin-releasing hormone, corticotropin-releasing hormone, thyrotropin-releasing hormone, progesterone, and estradiol besides other growth factors such as the Insulin-like growth factor 1 and 2 (IGF-I and IGF-II) [32].

Inherent to its secretory phenotype, the placenta is also a target of hormonal biological activity because it expresses most receptors for these hormones and growth factors. Therefore, placental hormones act in endocrine, paracrine, and autocrine pathways in the Maternal-placental-fetal-unit (MPFU).

In the context of GDM, diverse hormonal actors take part in the beginning or progression of the hyperglycemic, insulin resistance, oxidative stress, and the meta-inflammatory state of this disease. Three years ago, Madhusmita Rout and Sajitha Lulu proposed a network of maternal and placental genes and their potential impact on the transport of nutrients from mothers with GDM to their babies [33]. Through diverse in silico tools analyzing the interaction of gene/protein/miRNA/transcription factors, they described an important dysregulation of certain placental hormones, including leptin, insulin/IGF-I and their receptors, and the placental growth hormone receptor. Recently, other hormonal candidates that might play a role in the pathophysiology of GDM have been described through comprehensive bioinformatics, gene analysis, ROC analysis, and RT-PCR, including β-hCG, oxytocin receptors, binding proteins of IGF-I, and some cytochromes involved in the steroidogenic pathway [34,35].

### 2.1. The Insulin and IGF-I/IGF-II Axis and Molecular Pathways Primarily Disturbed in GDM Placentae

Insulin resistance is one of the first alterations in the pathogenesis of GDM. Insulin acts through two known receptor isoforms: insulin receptor A (IR-A) and insulin receptor B (IR-B). These are heterodimer receptors composed of an extracellular α-subunit that binds to insulin, and an intracellular β-subunit that binds to the insulin receptor substrate 1 (IRS-1). Both isoforms are transcripts of the same *INS* gene. However, the IR-A product lacks a sequence of 36 nucleotides in the C-terminal of the α-subunit as a result of alternative splicing of exon 11. In contrast, IR-B is the result of the full transcription of the *INS* gene [36,37]; this explains the differences in their activities. Insulin has a higher affinity for IR-A than for IR-B. In addition, IR-A activates the Ras-Raf-mitogen-activated protein kinase (MAPK) cascade, whereas IR-B activates the phosphatidylinositol 3-kinase (PI3K) and protein kinase B (Akt) signaling.

The study of the cellular pathways activated by insulin began around 33 years ago [38]. Since then, diverse research papers have scrutinized these cellular cascades, and finally, in 1995, the role of PI3K and Akt in the glycemic control and other metabolic activities of insulin was described [39,40]. After IR-B binding to insulin, PI3K binds to tyrosine-phosphorylated IRS proteins, leading to the formation of phosphatidylinositol (3,4,5)-triphosphate (PIP3). Downstream effects of PIP3 lead to activation of 3-phosphoinositide dependent protein kinase (PDK)1 and the subsequent activation of a variety of kinases, of which Akt 1–3 is the best-studied [41]. Main metabolic activities of insulin are related to Akt phosphorylation: (i) increased glycogen synthesis by inactivation of glycogen synthase kinase-3 (GSK3) α/β and activation of glycogen synthase [39]; (ii) Decreased transcription of gluconeogenic genes in liver and autophagy genes in muscle, by phosphorylation of forkhead box (FOX) transcription factors [42]; (iii) stimulated protein synthesis and suppression of autophagy by phosphorylation of tuberous sclerosis 2 (TSC2) and the 40 kDa proline-rich Akt substrate (PRAS40) which leads to activation of mTORC1 [43,44]; (iv) increased glucose uptake by phosphorylation of TBC1 domain family member 1/Akt substrate of 160 kDa (TBC1D4/AS160) which regulates trafficking and translocation of GLUT4 cytoplasmic vesicles to the plasma membrane [45]. In line with their metabolic effects, IR-B is preferentially expressed by classical insulin-sensitive cells such as hepatocytes, adipocytes, and myocytes, which have important roles in glucose, lipid, and protein metabolism.

On the other hand, activation of IR-A induces the Grb-2/Erk 1/2 MAPK pathway related to cell growth, differentiation, and survival processes [46]. This isoform is predominantly expressed in cancer tissues, the brain, hematopoietic cells, and the placenta [36].

Although the human placenta expresses both insulin receptors, IR-A is expressed at higher levels than IR-B [47]. This differential ratio is probably related to the need for tight control of the crucial proliferative and pro-differentiation pathways during pregnancy. On the other hand, there are redundant placental pathways to help in the vital fetoplacental glucose transfer, besides insulin/IR-B mediated GLUT-4 translocation [48,49], as occurs in insulin-dependent tissues. During the first trimester of pregnancy, IR-A is expressed more in the apical membrane of syncytiotrophoblasts whereas at term IR-B is concentrated in endothelial cells of the villi microvasculature [50].

In Figure 1, we summarize the disturbed molecular pathways in GDM placentae. Immunohistochemical and blotting studies showed that GDM placentae have a lower total protein expression of IR-A, PIP3, and IRS-1 in comparison to placentae from uncomplicated pregnancies. Interestingly this occurs independently of the metabolic control of the disease which implies profound and sustained effects in placenta signaling networks [51,52]. Also, it seems that the obesogenic environment is an additional regulating factor of insulin signaling among GDM placentae, involving IRS-2, PI3K, and GLUT4, the most sensitive targets to obesity [53]. Additionally, it has also been reported that GDM placentae have a pronounced phosphorylated pattern of IR and IRS-1 proteins, accompanied by hyperphosphorylation of STAT-3, MAPK 1-3 (Erk 1/2), and Akt [54,55]. Altogether, these data suggest that GDM primes the placenta to overstimulate the insulin signaling to compensate sustained exposure to hyperglycemia. The deficit in the total protein levels of these mediators results in insufficient placental glucose uptake, and consequently in a hyperglycemic state [56,57]. Although, other authors did not observe differences in the total protein levels of placental IR and IRS-1 [54,58]. Considering these inconsistencies, we believe more studies are needed to clarify insulin signaling in GDM placentae and to understand how placental imbalance in these signaling pathways results in higher levels of inflammatory cytokines, adipokines and oxidative reactive species, insulin resistance, and vascular disorders, all of which prevail in the local placenta and peripheral tissues of GDM mothers.

Another system that can cross-react with insulin signaling is the IGF-I axis. The system consists of two ligands IGF-I and IGF-II, two receptors IGF-1R and IGF-2R, six IGF binding proteins (IGFBP 1–6), and four insulin-like growth factor binding protein-related peptides IGFBP-rP1-4 [55,59]. These growth factors mainly regulate growth and metabolism throughout the life cycle, but its activity is critical during intrauterine life for mammalian development. They provide a signal to cells to indicate that adequate nutrients are available, and therefore to enhance cellular protein synthesis, to favor hypertrophy, and to stimulate cell division [60].

Insulin and IGF-I present a relatively low homology of 34% (BLAST alignment: CAA40342.1 and AAN39451.1), but their receptors are highly homologous, around 57% (BLAST alignment: CAA28030.1 and AAA59452.1). These similarities in their structures generate promiscuous interactions between them. IGF-II binds to IR-A with an affinity close to that of insulin, but it does not bind to IR-B [61]. Additionally, random hybrids of IR-A/IR-B and hybrids of IRs/IGF-1R have been reported in the placenta [36,62,63]. IGF-1R/IR hybrids bind IGF-I and IGF-II with high affinity but bind insulin with a relatively low affinity [64].

Given the structural similarities between components of the IGFs/Insulin axis, it is not surprising that these hormones share signaling pathways. IGF-I binds to IGF-1R and activates two main cascades: (i) PI3K/Akt pathway via IRS-1 phosphorylation which predominantly leads to metabolic effects; (ii) Ras-Raf-MAPK pathway via SHC domain proteins which control cellular growth and differentiation [59]. On the other hand, IGF-II binds with high affinity to IGF-2R, also known as the cation-independent mannose 6 phosphate receptor. This interaction targets IGF-II for its lysosomal degradation and consequently, IGF-2R sequesters IGF-II, controlling the circulating levels of this hormone. Therefore, the biological activity of IGF-II is exclusively derived from its binding to IR-A or IGF-1R, considering that it does not bind to IR-B, as mentioned before [65].

The placenta synthesizes all components of the IGF axis from early stages at 7 weeks of gestation. IGF-I was more expressed in the second and third trimesters of pregnancy in comparison with early pregnancy, and it was expressed by practically all placental cells except syncytiotrophoblasts. Whereas IGF-II was profusely expressed by cytotrophoblasts, mesoderm core, basal plate, columnar cytotrophoblasts, amnion, and chorion. IGF-IR was expressed ubiquitously in the placenta, except in Hofbauer cells. All six IGFBPs were expressed in decidua basalis and parietalis [66].

One key insulin-like action of IGF-I is related to glucose metabolism [67]. Biomedical and clinical studies indicate that IGF-I is a hypoglycemic factor that increases glucose uptake in different kinds of cells, including euglycemic trophoblasts [68,69,70]. However, deeper studies are needed to confirm its metabolic effects in the GDM milieu. In contrast, IGF-II seems to present a hyperglycemic effect since overexpression of IGF-II in pancreatic β-cells results in the development of T2DM [71].

Excessive fetal growth and weight is a common complication from GDM newborn babies. Macrosomia has been explained by two central modulators: hyperglycemia and activation of the IGF-I axis. Maternal hyperglycemia increases energetic substrate availability and then stimulates excessive growth and adiposity in GDM mothers [72]. In fact, an increased concentration of glucose transporters GLUT1 and GLUT9 has been observed in GDM placentae, which favors an increased placental and fetal D-glucose uptake [49,52,73,74] (see Figure 1).

Concerning IGF-I signaling, different serum components of the pathway have been measured in GDM patients in the last 20 years. In a recent meta-analysis developed by Dr. Wang’s group, in which they analyzed 12 independent studies, they found GDM was consistently associated with higher maternal IGF-I levels in mid-gestation (20–29 weeks) and late-gestation (>30 weeks), whereas serum IGF-II did not present significant changes between GDM and control mothers [60]. Interestingly, most data show significantly lower cord levels of IGFBP1, IGFBP2, IGFBP3, IGFBP-6 or IGFBP-rP1 [55,75,76], and lower maternal levels of IGFBP1 and IGFBP2 [55]. All IGFBPs bind both IGF-I and IGF-II with similar affinities (except IGFBP-6 which is essentially IGF-II specific [77]). Their metabolic effects are related to inhibition of IGF-I signaling by sequestrating it into a circulation reservoir. Consequently, diminished levels of IGFBPs and IGFBP-rPs result in higher cord blood levels of free-IGF-I in GDM patients [75]. Further, the research group of Dr. Sciacca identified an increased phosphorylation pattern of IGF-1R in placentae from metabolic uncontrolled mothers with GDM and T2DM [58]. These changes support a persistently activated IGF-I signaling in GDM placentae by increased activity of free-IGF-I (see Figure 1).

It is well known that the growth hormone (GH)-IGF-I axis is the major regulator of longitudinal growth along life. In addition, significant and positive correlations between the birth weight of newborns from GDM mothers and maternal serum IGF-I or molecules of the IGF-I signaling in GDM placentae have been described [52,78,79]. Therefore, overactivation of IGF-I signaling may be one critical factor involved in the development of macrosomia in babies from GDM mothers. Other morphologic changes in the placenta are also related to macrosomia, including broader intervillous spaces, increased terminal villus volume, a large proportion of immature villi, and a larger syncytiotrophoblast surface allowing higher amounts of glucose to cross the placenta [20,21].

One additional hypothesis in excessive fetal weight gain in GDM is related to IR/IGF-1R hybrids. A high proportion of these hybrids has been reported in skeletal muscle and adipose tissue of T2DM patients [63,80,81], and in placentae from insulin-resistant women [82]. IR/IGF-1R hybrids increase the binding sites for IGF-I and IGF-II, favoring IGF signaling, proliferation, and anabolic processes, as has been previously published in cancer models [83]. However, this hypothesis, which deserves to be further explored, has not so far been studied in the placentae of mothers with GDM.

### 2.2. The Role of Pancreatic β-Cells and Lactotroph Hormones in GDM

In a healthy pregnancy, insulin resistance increases between 50% and 60% in the third trimester, compared to the pre-pregnancy period [84]. This physiologic resistance is needed to ensure adequate delivery of glucose to a fast-growing fetus. In response, the mother needs to expand her capacity for insulin secretion which is achieved by an increase in β-pancreatic cell mass and number, finally leading to a euglycemic pregnancy. The lactotroph hormones prolactin (PRL) and human placental lactogens participate in cell-specific β-cell responses to counteract the physiological insulin resistance developed during pregnancy. These pancreatic adaptations occur before the onset of insulin resistance in pregnancy [9].

The maternal decidua is the main extra-pituitary source of PRL synthesis [85,86], although columnar trophoblasts and villous cytotrophoblasts of the placenta can also synthesize it to a lesser extent [85,87]. On the other hand, placental lactogens are exclusively synthesized during pregnancy by fetal syncytiotrophoblasts and include human placental lactogen (hPL), human chorionic somatomammotropin A and B (hCS-A and hCS-B), and the placental growth hormone (PGH). Evolutionary studies indicate that placental lactogens are closely related in their chemical structure to the human Growth Hormone (GH) as a result of three duplications and one deletion in the *GH* gene [88]. Derived from this structural homology, lactogens share with GH their binding capacity to both somatogenic and lactogenic receptors [89]. GH binds primordially to the hGH receptor and acts as a somatogen, whereas PRL and hPL bind to the prolactin receptor (PRL-R) and act as lactogens. PRL-R is a member of the cytokine receptor superfamily which presents 3 structural regions: an extracellular ligand-binding domain, a hydrophobic transmembrane domain, and an intracellular signaling domain. Multiple promoters and alternative splicing of the *PRLR* gene generate several isoforms which vary exclusively in their intracellular domains and potential recruitment of signaling mediators [90]. After ligand binding, PRL-R dimerizes which leads to the trans-phosphorylation of tyrosine residues present in Janus kinase 2 (JaK-2). This is followed by recruitment of signal transducers and transcription activators (STATs) -1, -3, or -5 which dimerize and migrate to the nucleus to enhance the expression of PRL-dependent genes [91]. During pregnancy, the binding of hPL or PRL to the long isoform of PRL-R in pancreatic β-cells activates the JaK-2/STAT-5 pathway which results in metabolic adaptations of these cells characterized for higher transcription of GLUT-2, glucokinase, insulin, survivin, cyclin D2, and Bcl6 genes [92]. GLUT-2 favors glucose uptake by β-cells, then glucose is phosphorylated by glucokinase and enters glycolysis/Krebs cycle/oxidative phosphorylation to increase ATP production. Higher ATP/ADP ratio blocks ATP-sensitive potassium channels, K^+^ accumulation depolarizes β-cells, and voltage-gated calcium channels become activated. The resultant rise in intracellular Ca^2+^ triggers insulin secretion [93]. Additionally, transcription of survivin, cyclin D2 and Bcl6 genes increases cell mitotic divisions, avoids apoptosis, and stimulates the expansion of pancreatic islets during normal pregnancy [94]. Beta-cell proliferation is also dependent on the downstream serotoninergic effect of both, PRL and hPL [95]. hPL is more potent than PRL to increase insulin secretion and β-cells proliferation, whereas GH has lower potency [96,97]. We now know that the morphologic changes in pancreatic β-cells related to pregnancy occurs largely through hPL and PRL action, but Hepatic Growth Factor (HGF), Epidermal Growth Factor (EGF), vitamin D, progestins and estrogens are also implicated [98,99,100,101] (see Figure 2).

Pregnancy is considered a physiologically hyperprolactinemic state, in where PRL potentiates glucose-stimulated insulin secretion and β-cell mass. However, exacerbated hyperprolactinemia, as occurs with a prolactinoma, is related to insulin resistance [102]. In GDM, blood levels of hPL are higher than in normal pregnancies and correlate with increased placental weight, macrosomia, hyperglycemia, insulin resistance, and altered values in an Oral Glucose Tolerance Test (OGTT) [103,104]. In contrast, GDM mothers present similar or even lower levels of PRL than normal pregnant women [105]. This relative contradiction seems to indicate that a delicate balance of PRL and hPL during pregnancy is needed to achieve adequate pancreatic β-cells proliferation and to avoid insulin resistance [106]. In a diabetic mouse model, low- and high- PRL treatment induced β-cell proliferation; however, low PRL levels reduced hepatic insulin resistance whereas high PRL exacerbated it and elevated apoptosis of β-cells [107]. A DNA sequencing study also supports the critical role of PRL-R in the control of glucose metabolism in GDM patients. In particular, two single nucleotide polymorphisms in the *PRLR* gene were associated with a 2-fold risk for developing GDM [103]. The study of physiological control of pancreatic β-mass expansion by lactotroph hormones still needs to be enlarged in the context of GDM and their co-interactions with estrogens/progestins hormones, inflammation, and obesity.

## 3. Hyperglycemia-Induced Metainflammation in GDM Alters Placental Immune Cells Population Favoring an Inflammatory Cytokine Signature and an Imbalance in Adipokines and Defense Peptides

The persistent exposure to hyperglycemia in GDM mothers results in a systemic response of inflammation named metainflammation [108,109]. This term defines a sustained low-grade inflammatory state characterized by an increase in serum levels of pro-inflammatory cytokines and tissue macrophage infiltration in the absence of tissue damage. Metainflammation in the context of GDM is favored by metabolic disorders such as maternal obesity or excessive weight gain, which induce inflammatory pathways leading to insulin resistance [110]. This persistent inflammatory state in GDM mothers has been associated with an increased risk for diabetes, obesity, and other poor outcomes and diseases in their offspring [111,112,113,114].

Several clinical studies in GDM patients have demonstrated alterations in a broad profile of inflammatory mediators, including lower serum levels of anti-inflammatory interleukin (IL)-10 and adiponectin, higher serum levels of pro-inflammatory TNF-α and IL-6, increased maternal serum adipokines (chemerin, leptin, omentin, visfatin, and the fatty acid-binding protein 4 FABP-4), Th1 cytokines (INF-γ, IL-2, IL-18) and chemokines (CXCL16, IL-8) [115,116,117,118,119,120]. Intracellular signaling of these inflammatory mediators is related to a worse fetal prognosis, such as preterm delivery and premature rupture of membranes [121].

### 3.1. The Role of Immune Cells in GDM Placentae

It is well known that hyperglycemia affects innate immunity and that there is a diabetes mellitus-dependent vulnerability to infections [122]. This scenario affects the correct function of innate immune cells as well as Toll-like receptor (TLR)–dependent responses at the MPFU, including alterations in the immune functions of monocytes/macrophages, dendritic cells, NK cells, granulocytes, and Hofbauer cells. 

Similar to what is seen in adipose tissue, obesity and GDM predispose to macrophage, granulocyte, and T lymphocyte infiltration into the placenta [120,123,124]. Indeed, the placenta expresses mRNA for CD68, CD14, EMR-1, and TCRa (immune cell infiltration markers), which are known to be increased in GDM placentae [123,124]. Macrophages infiltrated in GDM placentae present an M1 phenotype with a strong inflammatory response characterized by high expression of IL-6, TNF-α, IL-1β, IL-8, and the monocyte chemoattractant protein 1 (MCP-1) [120,123]. 

Hofbauer cells are fetal macrophages immersed in placental villous and present an anti-inflammatory profile (M2). In a rat model of GDM, Hofbauer cells switched their M2 profile towards M1 (pro-inflammatory) and induced the oxidative stress pathway [125]. In this regard, Hofbauer cells when treated with high glucose switch their profile to an M1-type, triggering inflammatory pathways [125], similarly to macrophages in a high-glucose environment [126]. However, in human placenta from GDM women, Hofbauer cells seem to preserve their M2 phenotype despite a hyperglycemic environment [127]. M2 macrophages play a relevant role in tissue remodeling during placental development, so further studies are needed to understand the effect of GDM on these cells.

Neutrophil activity is also altered in GDM. Neutrophils seem to be significantly activated, forming a high number of neutrophil extracellular traps (NETs). Under normal circumstances, these web-like structures represent an additional mechanism of the innate immune system to protect us from invading microorganisms. However, in pathological conditions such as diabetes and cancer, platelets may also get trapped, contributing to the pathological effects of NETs, which include damage to the endothelium and thrombotic events. Notably, numerous neutrophil infiltrates have been detected in the placentae of GDM-patients [128]. In addition, elevated neutrophil-derived products including nucleosomes, neutrophil elastase, and free DNA have been found in the plasma of diabetes mellitus patients [129]. Elevated first-trimester neutrophil count has also been associated with the development of GDM and adverse pregnancy outcomes [130]. In the placenta, extracellular traps formation may be triggered by infection (e.g., by bacteria or their products) or by inflammation (e.g., preeclampsia and GDM). Notably, extracellular traps comprise a vast array of molecules with antimicrobial activity, such as elastase, cathepsin G, defensins, myeloperoxidase, hCTD, and bacterial permeability-increasing protein, which explain their bactericidal effect [131]. However, the involvement of these immune structures in some noninfectious, autoimmune, and inflammatory processes grants further studies to understand their participation in GDM pathophysiology.

Regarding NK cells, diverse comprehensive reviews have described the critical participation of NK cells, particularly decidual NK cells with CD56bright/superbright and CD16- phenotype, in pregnancy development [132,133]. However, few original articles have revised the role of these cells in GDM pathophysiology. A higher percentage of cytotoxic NK cells (CD16+ CD56dim) was observed in maternal serum of overweight GDM patients and placental extravillous tissue in comparison with euglycemic women [119,134].

### 3.2. Major Placental Cytokines in GDM

The placenta presents not only an active endocrine function, but also an important immune-modulatory action characterized by the synthesis of diverse cytokines, chemokines, and adipokines, as well as their receptors. The implication of pro-inflammatory cytokines in GDM pathology has been demonstrated in numerous studies [135,136,137,138]. Gene microarray experiments in GDM placentae showed increased expression of genes for stress-activated and inflammatory responses, with upregulation of interleukins, leptin, and TNF-α receptors and their downstream molecular adaptors [139].

Undoubtedly, the signaling of NFκB is the main regulator of inflammatory pathways in normal and GDM placentae. After TNF-α binding with their receptor TNFR1, the adaptor protein TRADD is recruited and associated with the death domain of TNFR1. TRADD acts like platform binding for TRAF2 and RIP adaptor proteins which eventually activate the TAK1 kinase to phosphorylate and activate the IKK complex formed by the catalytic subunit IKKα and IKKβ, and the regulatory subunit NEMO. The IKK complex phosphorylates the IκB proteins that are constitutively bound to NFκB, keeping this factor in the cytosol. The serine phosphorylation of IκB proteins promotes their ubiquitination and proteolytic degradation by the proteasome, free allowing the nuclear translocation of NFκB [140,141]. The NFκB upregulates target genes that encoded pro-inflammatory cytokines, inducing a chronic inflammatory loop that contributes to the development of insulin resistance. 

NFκB can be activated by endogenous molecules released during tissue damage and oxidative stress, including debris from apoptotic, saturated fatty acids, heat shock proteins, advanced glycation products (AGEs) which are recognized by the TLR-4 receptor [142]. In GDM and maternal hyperglycemia there is a positive association with an increase of TLR-4 and NFκB signaling in the placenta [143,144]. TLR-4-induction of NFκB signaling in the placenta is an important mechanism that is altered during gestational diabetes; however, further studies are needed to elucidate the involvement of innate immunity in trophoblast functionality.

Clinical clamp assays and in vitro studies of placental perfusion demonstrate that TNF-α is the most significant independent predictor of insulin sensitivity in GDM patients, indicating close crosstalk between the immune and endocrine axis [145]. This immune interaction was corroborated in clinical association studies of GDM/obesity and maternal circulating levels of TNF-α and IL-6; this positive association remained after adjustment for total adipose mass [136,146,147,148,149]. Currently, it is known that cytokines such as TNF-α and IL-6 favor insulin resistance through inhibition of the insulin signaling cascade. Activation of the c-Jun N-terminal kinase (JNK) and IκB kinase (IKK) pathway targeted by TNF-α, increases serine phosphorylation of the IRS-1 and blocks normal tyrosine kinase activity of the IR [150]. On the other hand, IL-6 can activate the mammalian target of rapamycin (mTOR) which phosphorylates IRS-1 in serine residues, in a similar manner as TNF-α. Another molecular signaling of insulin resistance mediated by IL-6 is related to STAT-3 activation and the consequent activation of the Suppressor of Cytokine signaling (SOCS), which inhibits tyrosine phosphorylation of IRS-1. Both serine phosphorylation and blocked tyrosine phosphorylation of IRS-1 by inflammatory mediators impaired insulin action and subsequent insulin-regulated glucose uptake [151].

IL-1β is another important inflammatory mediator that has been shown to be increased in the placentae of obese mothers and with GDM [119,152,153]. In a first-trimester trophoblast cell line (Sw.71), high glucose levels (25 mM) increased trophoblast production of uric acid, which activated the inflammasome pathway, and positively regulated IL-1β release [154]. In mouse models for GDM, an increase in uterine and placental IL-1β levels and impaired glucose tolerance has been observed. In contrast, treatment with an anti-IL-1β antibody improved glucose tolerance in GDM mice [155]. Results obtained from in vitro models of hyperglycemia support the clinical data of the inflammatory environment in the placenta of women with GDM and provide evidence of the consequences that this generates on the anatomy and functionality of the placenta. Exposure of placental trophoblast cells to high concentrations of glucose (25 and 50 mM) significantly induced the secretion of cytokines and chemokines such as TNF-α, IL-1β, IL-6, IL-8, GRO-α, RANTES, and G-CSF [154]. Furthermore, high glucose concentrations suppress trophoblast viability and proliferation in vitro. These effects were mediated by the increase in miR-137, which in turn decreases the expression of the protein kinase activated by AMP (PRKAA1), positively regulating the placental secretion of IL-6 [156], and upregulating Bax/Bcl-2, COX-2, and caspase 9 expression [157].

It is well known that the migration of extravillous trophoblast into the maternal decidua and their interaction with endothelial cells from uterine spiral arteries are critical processes in placental development. In this sense, in vitro studies support an active intercommunication between trophoblasts and endothelial cells. The first-trimester trophoblast exposed to hyperglycemia (25 mM) induces the release of exosomes which in turn induces the endothelial release of IL-4, IL-6, IL-8, INF-γ, and TNF-α [158]. Together, the placental alteration of matrix metalloproteases, pro-angiogenic, and anti-angiogenic factors, as well as the shift towards a pro-inflammatory environment can explain the peripheral endothelial dysfunction in GDM [139,159,160]. Also, this endothelial imbalance may help to explain the observed high vasculogenesis rate, and capillary immaturity in GDM placentae [20,21], as will be discussed later in this review.

### 3.3. Altered Production of Adipokines by the GDM Placenta

Adipokines are bioactive polypeptides whose dysfunction is involved in inflammation, obesity, insulin resistance, and cardiovascular diseases [161]. Although adipokines have been described as secretion products of adipose tissue, the placenta can synthesize adipokines such as chemerin, omentin-1, visfatin, leptin, and adiponectin-like adipose tissue [8,135].

Leptin acts as a regulator of satiety and energy expenditure in the central nervous system [162]. Maternal plasma leptin levels increase in the first and second trimester of pregnancy and return to pre-pregnancy levels after delivery [163]. The placental synthesis of leptin is regulated by the different placental hormones such as β-hCG and estradiol [164,165]. Several studies have shown that the secretion of leptin by the placenta exerts autocrine actions stimulating the proliferation and survival of trophoblast cells [166]. Also, leptin increases placental lipid catabolism and vasodilation, possibly increasing the availability and transport of nutrients thus favoring fetal growth [167].

Regarding GDM, most researchers reported an increase in plasma and placental leptin levels [115,168,169,170] while other researchers did not observe a difference compared to healthy pregnancies [171]. In vitro evidence has shown that insulin induces leptin expression in trophoblast cells [162]. It has been found that maternal hyperglycemia in GDM regulates the leptin levels in umbilical cord blood generating macrosomia in the baby and increasing their risk of obesity in the future [168].

An increase of leptin and leptin receptor expression was found in the GDM placentae [172], contributing to the increased placental weight gain observed in GDM, along with the IGF-I axis as described before (see Figure 1). The binding of leptin with its receptor activates the signaling pathways MAPK, PI3K, and JaK-STAT which are also shared by IR [54]. In GDM placentae, the basal phosphorylation of STAT-3, MAPK 1/3, and Akt are increased, causing resistance to subsequent stimulation with Leptin or Insulin in vitro, suggesting crosstalk between insulin and leptin signaling in the human placenta [54].

In explants of the placenta, leptin significantly increases the release of IL-1β, IL-6, TNF-α, and prostaglandin E2 (PGE2) [173]. Similarly, leptin stimulates IL-6 secretion in trophoblast cells [174,175]. This increase in the production of pro-inflammatory cytokines evokes a chronic inflammatory milieu that in turn improves leptin release in the placenta, generating a vicious inflammatory loop [176].

Adiponectin is the second most studied adipokine. This peptide improves the IR signaling, makes lipid oxidation more efficient, inhibits the gluconeogenesis and the TNF-α signal in adipose tissue [177]. Adiponectin circulating levels increase during the first and second trimesters of normal pregnancy and later decrease post-partum [178]. Unlike leptin, low levels of adiponectin were found in maternal GDM serum compared to women with normal glucose tolerance [179,180]. It has been proposed that this decrease in serum adiponectin levels may serve as an early predictive factor for GDM development.

In terms of human placentae from pregnancies complicated with GDM, a significant downregulation of adiponectin mRNA and an upregulation of adiponectin receptor 1 (ADIPOR1) has been reported [181]. In the same study, it was observed that the cytokines INF-γ, TNF-α, and IL-6 differentially regulate the expression of adiponectin receptors (ADIPOR1 and ADIPOR2), as well as the expression and secretion of adiponectin. The researchers observed that placental adiponectin suppressed MAPK phosphorylation, particularly ERK1/2 and p38 that are essential for the onset of trophoblast differentiation, implantation, and placentation [181]. Furthermore, there is in vitro evidence that adiponectin promotes the trophoblast invasion by augmenting Matrix metalloproteinase (MMP)-2 and -9 production, and downregulating TIMP-2 mRNA expression [182]. Whereas adiponectin increases insulin sensitivity and modulates the invasion of trophoblasts, this adipokine could limit fetal and placental growth. Recent studies suggest that hypomethylation of the adiponectin gene in the placenta correlates with maternal insulin resistance and hyperglycemia, and with fetal macrosomia [183,184]. Secretion and expression of other placental adipokines have been described in women with GDM and obesity. Some studies showed an increase in many of these adipokines while others found no changes compared to women with normoglycemia [177]. More studies are needed to understand the impact of these adipokines on the development of GDM in the human.

### 3.4. Placental Innate Defense-Peptides in GDM

Regarding the innate immune system of the MPFU, the placenta, trophoblasts cells, decidual cells, stromal cells, and fetal membranes can produce several antimicrobial peptides, including human β-defensin (HBD)-1, HBD-2, HBD-3 and HBD-4, S100 proteins, human cathelicidin (hCTD) and human neutrophil peptides 1–4 (HNP 1–4) [185,186,187,188,189,190,191]. Moreover, the placenta also produces histones capable to neutralize certain bacterial endotoxins [192]. The main function of these defense peptides is to rapidly kill invader microorganisms; however, these multifunctional, amphipathic molecules also participate in angiogenesis, cell migration, and immune system modulation [193].

At this time, there is scarce information about the effect of diabetes and/or glucose levels on human placental antimicrobial peptides. In human amniotic epithelial cells, the high glucose culture medium is known to downregulate HBD2 production [194]. In other cell types such as macrophages, high glucose levels inhibit hCTD expression. However, in *Mycobacterium tuberculosis*-infected macrophages, hCTD levels increased as mycobacterial burden augmented, irrespective of the hyperglycemic environment [195]. Similarly, in a rat model of diabetes, lower levels of defensin BD1 have been found, but interestingly, they could be restored by insulin. On the contrary, defensin BD2 levels were found significantly higher than in non-diabetic animals, which was interpreted as the result of a glucose-dependent increased inflammatory state [196]. Likewise, in biopsies from diabetic foot ulcers, all studied defensins (HBD1-4) were overexpressed while hCTD was decreased in comparison to healthy skin. Remarkably, when an infectious agent was present, a significantly lower hCTD expression was observed. The authors concluded that the amount of antimicrobial peptide present in these diabetic tissues was not able to efficiently contain the infection [197].

Altogether, these results suggest that high blood glucose differentially regulates the expression of innate immune system components and that there seems to be an interaction between this hyperglycemic state and other assaults that may occur in diabetes, namely inflammation and infection. A particularly interesting case is that of hCTD, which is of primordial importance, especially for intracellular infections. In the human placenta, and similarly as in the cases just described, there is a differential inflammation-dependent regulation of hCTD and HBDs. Indeed, bacterial LPS endotoxin stimulated HBD2 and S100A9 mRNA levels, while significantly repressed basal and calcitriol-dependent hCTD expression [198].

## 4. Vitamin D Implications in Immune Regulation and Insulin Resistance in GDM

As discussed in the previous section, there is a major immune imbalance in GDM closely linked to the state of metainflammation. An important placental hormone with potent immune-modulating activity in pregnancy is calcitriol, the active form of Vitamin D (VD). Calcitriol shows a dual effect: anti-inflammatory (lowers inflammatory cytokines) and stimulator of the innate response (induces antimicrobial peptides expression) [189,198,199,200]. This suggests that the lower VD levels described in GDM [201] may worsen the inflammatory state and further decrease the innate immune response prevailing in this pathology. Specifically, hypovitaminosis-D may take its toll on placental hCTD, HBDs, and S100A9, which production depends on calcitriol transcriptional activity [189,198,202]. Moreover, maternal VD deficiency has been significantly associated with insulin resistance and a greater risk of GDM [203,204,205], which may be linked to the aberrant VD metabolism that takes place in the diabetic placenta [201,206]. Because of the latter, VD deficiency increases the susceptibility to maternal-fetal infections, most probably by the resultant limited immune response [207,208].

Importantly, the VD Receptor (VDR) is expressed in different organs/tissues, including the placenta and pancreas, and it is known to be involved in the regulation of glucose metabolism through modulating insulin production and secretion [101,201,209]. Indeed, VD-response elements are present in the human insulin receptor gene, which is activated by VDR-calcitriol [210,211]. Accordingly, recent reports support that calcitriol protects against insulin resistance and exacerbated inflammation at the MPFU in GDM [138,212]. Therefore, VD deficiency is associated with blood glucose and insulin alterations, partially explaining the suggested implication of VD deficiency as an independent risk factor for GDM [203]. 

## 5. Endovascular Changes in GDM: Endothelial Dysfunction and Overstimulation of Placental Angiogenesis

During the second trimester of pregnancy (20th week of pregnancy), the placenta receives 21% of the combined cardiac output; practically equal to the two principal organs receiving 20–25% of the cardiac output: the liver and kidneys [213,214,215]. This important blood influx directed to the placenta demands the formation of a highly efficient vascular network interconnecting the mother, the placenta, and the fetus. To satisfy this demand, the placenta must carry on two stages of vessel formation: vasculogenesis and angiogenesis. Vasculogenesis begins on day 21 after conception refers to the de novo formation of the primitive vascular plexus during embryonic development, via the differentiation of pluripotent hemangioblasts into endothelial cells. Then, angiogenesis takes place from day 32 until delivery, where the formation and remodeling of vascular trees occur through the proliferation and migration of endothelial cells from pre-existing blood vessels [216,217]. Both processes are essential for placental development as well as organogenesis during embryonic and fetal growth.

Physiologic control of angiogenesis is extremely complex and tightly regulated. It involves cellular interactions with the extracellular matrix, autocrine and paracrine signaling induced by hormones, cytokines, chemokines, matrix metalloproteinases (MMPs), miRNAs, as well as pro-angiogenic, anti-angiogenic factors and their receptors [218,219]. Among all these factors, VEGF (specially isoform A), known as the main inducer of angiogenesis both in vivo and in vitro [220], is actively induced by hypoxic environments through HIF-1α [221,222]. This pro-angiogenic molecule acts by binding to two membrane receptors: VEGFR1 (or Fms Related Receptor Tyrosine Kinase 1, Flt-1), and VEGFR2 (or Fetal Liver Kinase-1, Flk-1). Both receptors are highly expressed by terminal villous capillaries and the adjacent trophoblast [223,224].

Other important pro-angiogenic factors produced by placenta are the PlGF, which can also activate VEGFR1, neuropilin-1 (NRP-1), angiopoietins, fibroblast growth factor 2, platelet derived growth factor, and the insulin/IGF axis [225,226,227,228]. As mentioned before, HIF-1α is a major driver of VEGF transcription, however in vitro studies demonstrated PlGF is not regulated by this factor in the human placenta [229]. In trophoblast cells, PlGF expression decreases in low oxygen tension conditions, whereas normoxic conditions increase PlGF contrary to what occurs in other cell types [230].

The main anti-angiogenic molecules include soluble Flt-1 (sFlt-1), which antagonizes VEGF and PlGF by sequestering them and preventing their interaction with their VEGFR-1/-2 receptors, and soluble Endoglin (sEng), a TGF-β signaling antagonist. In vitro effects of sFlt1 include vasoconstriction and endothelial dysfunction, sEng amplifies the vascular damage mediated by sFlT1 and in trophoblast cells, sEng is released in response to hypoxia, oxidative stress or inflammation [231,232].

Since the placenta is a highly vascularized organ that supplies nutrients, growth factors, oxygen, hormones, and immune mediators to the fetus, all molecules of the angiogenic axis are tightly regulated throughout pregnancy. However, as mentioned before, clinical examination of GDM placentae shows increased villous vascularity often associated with thickened immature villi capillaries [20,21,24]. Diverse experimental evidence indicate that hyperglycemia promotes angiogenesis, vasoconstriction, and a higher vessel permeability [233,234,235,236,237,238]. Therefore, hyperglycemia in GDM has been highly associated with vascular damage, endothelial dysfunction, and aberrant overstimulated placental angiogenesis. All these changes result in severe endothelial damage, leading to impairment of both the maternal and fetal vascular system [239].

The role of VEGF axis in GDM is unclear. Some experiments showed a predominant pro-angiogenic balance characterized by higher serum levels or placental content of VEGF, PlGF, VEGFR2, and HIF-1α [159,240,241]. However, other researchers observed the opposite effect of these molecules on the VEGF axis in GDM placentae or trophoblasts under hyperglycemic condition [154,242,243]. In feto-placental vessels and capillaries of GDM placentae, decreased expression of VEGF has been found [244,245]. Despite this VEGF downregulation, feto-placental endothelial cells incubated with culture media of trophoblasts isolated from GDM placentae exhibit a more pronounced vessel network formation in vitro [159,160].

Therefore, it seems that other regulatory factors besides those involved in the VEGF/VEGFR axis may be participating in placental angiogenesis and should be considered to understand vascularization and blood perfusion in GDM placentae. These factors may be derived from syncytiotrophoblasts, mesenchymal stromal cells and/or endothelial cells. In addition, other non-classical players in the control of the angiogenic process in GDM patients have been evaluated. Dr. Lana Mc-Clements found GDM placentae have a decreased expression of the anti-angiogenic protein sirtuin 1 (SIRT-1), which may help to explain the increased vascularized network observed in GDM placentae. Additionally, they cultured a trophoblast cell line (obtained from first trimester/choriocarcinoma) under conditions of hyperglycemia (25 mM) and hypoxia (2.5 and 6.5% O_2_), and observed a downregulated expression of the anti-angiogenic proteins FK 506 binding protein-like (FKBLP) and SIRT-1 [24]. These results support the hypothesis that hyperglycemia and lower oxygen tension alter the balance of factors that regulate angiogenesis in the placenta, leading to vascular dysfunction and possibly the development of preeclampsia.

The research group of Dr. Mojgan Karimabad found increased serum levels of CXCL1 and CXCL12, chemokines with pro-angiogenic properties [246,247,248], in neonates from GDM mothers, whereas antiangiogenic chemokines CXCL9 and CXCL10 were downregulated, compared to neonates from healthy mothers without GDM [249]. In addition, upregulated expression and activity of MMP-2 and MMP-9 and downregulated expression of TIMP-2 has been observed in trophoblasts under hyperglycemic treatment (30 mM), favoring endothelial cell migration during angiogenesis [240]. Furthermore, the Membrane-type MMP1, a kind of MMP anchored to the cellular membrane and a key player in vascularization and angiogenesis, is also up-regulated in GDM placentae [250,251]. Interestingly, these researchers observed that this metalloproteinase is up-regulated by insulin, which signaling is overactivated in GDM placentae, as described earlier [54,55]. Also, the angiopoietin-related growth factor (AGF), another pro-angiogenic molecule, is elevated in serum of GDM mothers [252].

All considered, the changes in the angiogenic molecules profile in GDM placentae and mothers indicate an exacerbated placental vascularization, but with signs of endothelial cell dysfunction, higher vessel permeability, and compromised integrity of chorionic vasculature, like immature/injured villi capillaries. It has been hypothesized that the increased number of villi and vessels in pregnant women with hyperglycemia provides a greater surface for maternal-fetal exchange. This functional adaptation would facilitate the passage of glucose to the developing fetus, thereby contributing to explain fetal macrosomia [244,253].

Due to the human hematogenous placentation, fetal capillaries of placental villi are particularly vulnerable to any alteration in GDM maternal blood [254]. Diverse vasoactive agents are increased in GDM maternal serum, including proinflammatory cytokines, oxygen-free radicals, advanced glycation end products (AGEs), hyperglycemia, hyperinsulinemia, and hypoxia. Currently, it is well known that hyperglycemia and TNF-α actively stimulate plasma and endothelial production of reactive oxygen species (ROS) and toxic by-products of glycolysis, leading to the formation of AGEs [255,256,257] which damages macro- and especially micro-vasculature and may contribute to thrombotic and atherosclerotic events [257,258]. In addition, it has been hypothesized that this endothelial damage evidenced by increased angiogenesis rate and chorionic villous branching may conduce to increased peripheral vascular resistance. This may explain the higher maternal blood pressure and preeclampsia [24,239], that are highly frequent complications in GDM pregnancies [1,7].

## 6. GDM and Oxidative Stress

Disruption of the delicate equilibrium between antioxidants and pro-oxidants in the cellular and tissular milieu can create oxidative distress that induces accumulative damage by modifying the state of macromolecules such as proteins, lipids, and DNA [259]. A distressed intracellular and extracellular environment leads to unspecific oxidation of proteins and altered response patterns, as well as irreversible damage causing growth arrest, cell death, and inflammation [260].

Reactive Oxygen Species (ROS) are described as free radical highly reactive molecules derived from the reduction of molecular oxygen that are formed by reduction-oxidation (redox) reaction or by electronic excitation [259,260]. Some of the most known species are superoxide (O_2_^−^•), hydroxide (OH^−^•), and hydrogen peroxide (H_2_O_2_) [261]. ROS have a role in cell signaling including apoptosis, gene expression, activation of cell signaling cascades, and serving as both intra- and intercellular messengers [262].

The placenta, being a “new” organ, must establish multiple adaptations to a high oxygen-rich environment. In this scenario, the intervillous space is exposed to hypoxic conditions in the first 10–12 weeks of pregnancy, when the trophoblast plug in spiral arteries is removed or disintegrated and begins the entry of complete blood into intervillous space [263,264]. When the maternal blood circulation is open through the placenta, there is a significant increase in oxidative stress in trophoblasts. The gradual opening of increasing numbers of maternal vessels allows the placental tissue to adapt to the increased oxygen tension and oxidative stress. Some degree of placental oxidative stress occurs in all pregnancies towards term, however, under normal conditions, there are local compensatory mechanisms such as the rise of antioxidant’s glutathione peroxidase and catalase activity [265].

ROS generation by oxidative phosphorylation at the mitochondrial membranes may be pertinent to tissues with a high-energy demand or those containing large amounts of mitochondria such as the placenta [266]. Under normal physiological conditions, ROS play a crucial role in normal embryonic development and cell function, such as trophoblast invasion and vascular development in the placenta [265,267,268].

In the placenta, fluctuating oxygen conditions can increase ROS production, where they can act as signaling molecules. O_2_^−^• is produced via an enzymatic and non-enzymatic process that takes place mainly in mitochondria. Along a non-enzymatic process, leakage of electrons from enzymes of the mitochondrial respiratory chain takes place and reduces molecular oxygen, thereby forming O_2_^−^•.

As compensatory physiological mechanisms, the enzyme manganese superoxide dismutase (MnSOD) catalyzes the diminution of O_2_^−^• and the formation of hydrogen peroxide, which is transformed to oxygen and water by glutathione peroxidase and catalase [269,270]. Likewise, MnSOD catalase and other enzymes such as glutathione peroxidase and copper/zinc superoxide dismutase (SOD1) are present in the placenta acting as antioxidants.

### ROS Changes Associated with a Hyperglycemic Environment

Oxidative stress has been associated with multiple pathological conditions, including GDM [271]. Inflammatory modulators, such as TNF-α, have deleterious effects during gestation and especially in hyperglycemic environments by increasing the expression and activation of ROS precursors, like NADPH oxidase 4 (NOX 4) [272]. Experimental and clinical evidence support the idea that a hyperglycemic environment is associated with increased oxidative stress. It has been reported that GDM women have a decreased ability to compensate for oxidative stress, and this was associated with increased insulin resistance and reduced insulin secretion [273]. There is evidence suggesting that women with GDM produce high levels of free radicals and have compromised free-radical scavenging mechanisms. Increased ROS, together with the impaired peripheral antioxidant activity is related to the induction of congenital malformation in pre-gestational diabetic pregnancies plasma of women with GDM [274,275,276,277]. In vitro studies done in 3T3-L1 adipocytes support that oxidative stress triggers decreased GLUT-4 expression, as a result of impaired binding of nuclear proteins to the insulin-responsive element in the GLUT-4 promotor [278]. Thus, oxidative stress adds to the factors related to impaired peripheral glucose metabolic control.

In the GDM placenta, there seems to be another picture. Not only is there a reported increase in oxidative stress and lipid peroxidation compared to normal pregnant women [275,276,279], but there also appears to be a concomitant increase in antioxidant enzyme activity that compensates for it [26,275,280,281]. The activated placental Nuclear Factor Erythroid 2-Related Factor 2 (Nrf2)/Antioxidant Response Element (ARE) pathway might have led to an increased expression of antioxidant enzymes SOD1, and catalase. This may be viewed as a protective mechanism in the placenta from the further onslaught of oxidative stress [282]. Another related pathway, the Nrf2/Kelch-like ECH-associated protein 1 (Keap2) pathway, also plays a crucial role in transcriptional activation of antioxidant defense genes and restoration of redox homeostasis [282]. One more antioxidant mechanism is related to apolipoprotein D (apo D), which has been observed to increase in the villous trophoblast and adventitia tunica around the large blood vessels in placental tissue from GDM [283]. Altogether, this data suggests that placentae with GDM are more protected against oxidative damage but are more susceptible to nitrosative damage as compared to normal placentae. In Table 1, we review the main clinical results in oxidative status reported in GDM women.

Finally, recent evidence suggests that the cerebral cortex and hippocampus of male and female GDM offspring present an increase in ROS and lipid peroxidation, a disruption in the glutathione status, and decreased activity of catalase and SOD1; thus, they present a marked neural pro-oxidative environment. The researchers hypothesized that cognitive behavior could be modified in an age- and sex-dependent manner [299].

## 7. Adverse Perinatal Outcomes Related to GDM

As result of the metabolic alterations that occur in the mother, and particularly in the placenta during GDM, the mother is at higher risk of complications than healthy pregnant women. Obesity and excessive weigh gain are common conditions that increase GDM development risk [300]. Obesity per se is related to increased risk of pregnancy-induced hypertension, preeclampsia, risk of venous embolism, increased need for labor induction, and cesarean sections [301]. This is particularly important because of the higher rate of overweight and obesity in women of reproductive age. However, the hallmark of GDM is maternal hyperglycemia, resulting in a broad spectrum of clinical consequences for the mother and fetus in both the short and long term. In general, the severity of complications is related to the earlier onset of GDM and correlates inversely with the degree of glycemic control [302].

For the mother, short-term obstetric complications include hypertension, preeclampsia, premature delivery, and cesarean section [303,304]. In the long-term, the mother is in a high likelihood of recurrence of GDM in successive pregnancies above 48% [305,306]. Also, a robust body of literature supports GDM hyperglycemia predisposes mothers to developing T2DM and cardiovascular diseases years after delivery [10,307,308,309,310,311]. Although the proportion of GDM women who develop T2DM is highly variable, a recent meta-analysis indicates a relative risk of 8-fold (95% IC: 6.5–10.6) [10]. Subsequently, this increases the chance of developing cardiovascular disease and metabolic syndrome up to threefold [308].

On the other hand, GDM is associated with the development of fetopathies. Maternal hyperglycemia during GDM generates fetal hypoxia, which can evoke birth asphyxia and fetal death. Clinical evidence supports that fetal hypoxia increases erythropoietin production, which is associated with the development of polycythemia, hyperbilirubinemia and neonatal icterus [4]. As has been described in the previous sections of the manuscript, hyperglycemia can generate fetal macrosomia, mainly in overweight or obese women before pregnancy. Macrosomia is defined as a newborn weight greater than 4000 g. Higher fetal weight is not only related to total body mass but also to an excessive fat mass/lean mass ratio, indicating higher adipose mass deposition [312,313]. An early predictor of macrosomia is increased abdominal circumference by ultrasound between weeks 20 to 24 of gestation [11]. The fetus absorbs glucose, stores it in the form of glycogen in the liver, and the excess is converted into visceral fat, increasing the fetal abdominal circumference. Moreover, fetal epicardial fat thickness has been proposed as an earlier marker (before 24 weeks) to screen GDM women [314]. Macrosomia and hyperglycemia predispose to obstetrics complications such as shoulder dystocia at birth, plexus-brachial injury, respiratory distress syndrome, as well as the need for instrumented delivery [7,315].

The fetal hyperinsulinemia induced by GDM impairs pulmonary surfactant synthesis, delaying lung maturation by 1 to 1.5 weeks, and predisposing the fetus to develop respiratory distress syndrome [316]. In the Hyperglycemia and Adverse Pregnancy Outcome (HAPO) study, a large multinational-racial and ethnically cohort was analyzed, demonstrating that high maternal glucose levels are associated with increased birth weight above the 90th percentile for gestational age, primary cesarean delivery, clinically diagnosed neonatal hypoglycemia, and cord-blood serum C-peptide level above the 90th percentile. Secondary outcomes were delivery before 37 weeks of gestation, shoulder dystocia or birth injury, need for intensive neonatal care, hyperbilirubinemia, and preeclampsia [2]. These results were supported by other studies [303,317]. In addition, adverse fetal cardiac patterns have also been evaluated in GDM neonates. Main cardiotocographic alterations include higher risk for hypoxia-related ZigZag patterns, late decelerations of the fetal heart ratio, and greater risk of fetal asphyxia [318].

GDM has a lower risk of congenital malformations because it develops after organogenesis, unlike pregestational T1DM or T2DM, where the maternal hyperglycemia acts as a teratogenic factor [319]. Although early-onset of GDM could lead to a slight increase in the rate of birth defects to 16–22 percent [320,321,322]. 

Finally, fetal programming in GDM has been studied in the context of DOHaD hypothesis (developmental origins of health and disease) to understand the long-term consequences of growing in a hyperglycemic environment during life in utero. Hyperglycemia and maternal obesity generate epigenetic changes in fetal programming, increasing the risk for obesity, T2DM, impaired glucose tolerance, insulin resistance, coronary heart disease, chronic arterial hypertension, dyslipidemia, metabolic syndrome, and some cancers in the future life of the newborn [3,323,324,325,326,327]. Interestingly, it has been observed that GDM effects in childhood are manifested after two years of age, such as an increase in BMI and hyperglycemia [328]. Regarding this, the HAPO follow-up study (HAPO-FUS), which included 4160 children aged 10–14 years, demonstrated that maternal hyperglycemia correlates positively with increased glucose and insulin resistance in childhood. Furthermore, hyperglycemia was associated with the increase of adiposity and metabolic distress [323]. These data confirm the hypothesis that GDM causes long-term metabolic alterations in the offspring and demonstrate the importance of timely diagnosis and treatment of this disease.

## 8. Conclusions

The functional interplay between the placenta and the maternal adipose tissue, glycemic control, and overall maternal metabolism, requires a highly controlled equilibrium that, if disturbed, may lead to GDM. Known risk factors for GDM include overweight/obesity, lack of physical activity, prediabetes, and genetic predisposition. In this pathology, maternal hyperglycemia, carbohydrate intolerance, dysfunction of beta-pancreatic and endothelial cells, as well as insulin resistance, disturb placental structure and functions. This disturbance favors a metainflammatory environment associated with increased production of inflammatory cytokines, adipokines, and oxidative reactive species, leading to an abnormal endocrine, immune and antioxidant phenotype. This maternal/placental immunoendocrine dysregulation affects the mother as well as the fetus’s health in the short and long term. The severity of complications relates to an earlier onset of GDM and correlates inversely with the degree of glycemic control. Therefore, it is of utmost importance to understand the pathophysiology of GDM to develop intervention and prevention strategies, including the orientation of pregnant women to eat healthier foods and exercising. In Figure 2, we propose a multi-organ scheme integrating the role of the placenta in the immunoendocrine environment dysregulated during GDM.

## Figures and Tables

**Figure 1 ijms-22-08087-f001:**
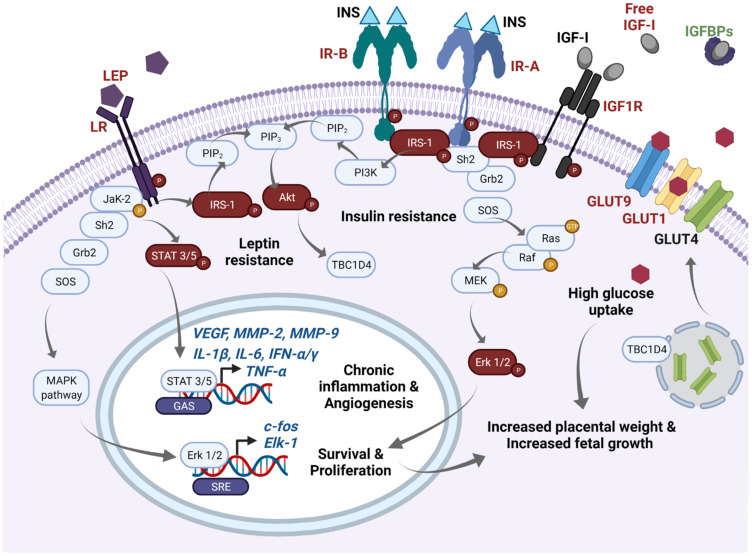
Main molecular pathways disturbed in GDM placentae. In this schematic, we highlight in red main molecules reported as overexpressed/overactivated in GDM placentae, whereas downregulation is highlighted in green. High expression of phospho-IR-A and phospho-IR-B has been reported in the GDM placenta. After insulin binding, IR-A phosphorylates IRS-1 and recruits Scheme 2. and Grb2 proteins which induce the GTPase activity of Ras, and then a GDP is exchanged by a GTP. This initiates a subsequent cascade of phosphorylations of Raf, MEK, and Erk 1/2. Finally, overexpressed phospho-Erk proteins translocate to the nucleus and recognize SRE sites favoring active transcription of c-fos and Elk-1. FOS proteins have been implicated as regulators of cell proliferation, survival, differentiation, and transformation. This pathway mediates increased placental weight and increased fetal growth in GDM. Additionally, increased levels of free IGF-I resulting from low IGFBPs serum concentrations also activate IR-A signaling as well as IGF1R and are related to proliferative effects. IGF-I excess has been also implicated in macrosomia and excessive placental growth in GDM women. On the other hand, after insulin binding to IR-B, phospho-IRS-1 activates PI3K and leads the formation of PIP3 from PIP2. Then, PIP3 activates Akt which mediates diverse metabolic effects; one of them includes GLUT4 translocation from endosomes to the cellular membrane through TBC1D4 signaling. High expression of GLUT1 and GLUT9, and probably GLUT4, mediates high glucose uptake in the GDM placenta, which can also participate in fetal and placental growth through an excess of this energetic substrate. Finally, GDM placentae present high expression of leptin and its receptor. High expression of phospho-leptin receptor recruits JaK-2 protein and activates STAT 3 or 5 proteins. Then, STATs dimerize and translocate to the nucleus and recognize GAS sites and provoke transcription of VEGF, MMP-2, MMP-9, TNF-α, IL-1α, IL-1β, IFN-α, IFN-γ, among others, which exacerbate the inflammatory placental milieu and contribute to stimulating the angiogenic process. Additionally, activation of the leptin receptor can also crosstalk with MAPK and Akt pathways. Sustained overactivation of all these pathways finally leads to clinical insulin and leptin resistance. GAS: Gamma-activated sequence. GDM: Gestational diabetes mellitus; GLUTs: Glucose transporters; IGFBPs: IGF-I binding proteins; IGF-I: Insulin-like Growth Factor 1; IGF1R: IGF-I receptor; INS: Insulin; IR-A: Insulin receptor type A; IR-B: Insulin receptor type B; LEP: Leptin; LR: Leptin receptor; SRE: Serum response elements.

**Figure 2 ijms-22-08087-f002:**
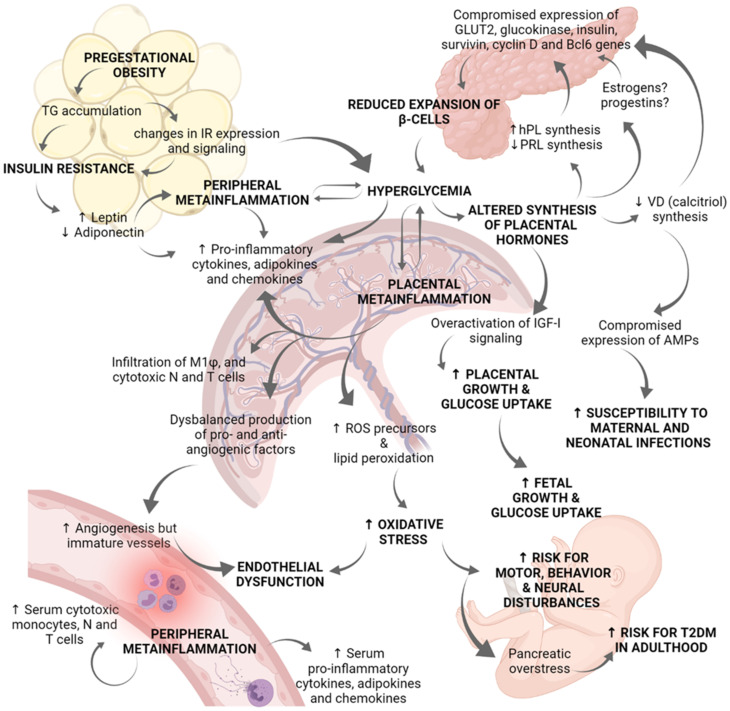
Role of placenta in the immunoendocrine dysregulations occurring in gestational diabetes mellitus. Hyperglycemia during pregnancy compromises the correct physiology of diverse organs, and particularly the placenta. Pregestational obesity is characterized by an excessive TG accumulation and changes in IR expression and signaling in adipose tissue, as well as in skeletal muscle cells. These changes in insulin-dependent tissues lead to insulin resistance. Then, adipose tissue secretes high levels of leptin and inhibits those of adiponectin. Leptin activation of JaK-2/STAT 3/5 pathway results in increased cytokine production contributing to peripheral metainflammation. Due to insulin resistance and altered IR signaling, women suffer chronic hyperglycemia. Clinical and biomedical studies indicate there is a positive regulatory loop between hyperglycemia and metainflammation, in which hyperglycemia induces placental and peripheral synthesis of pro-inflammatory cytokines, chemokines, and adipokines. Metainflammation also alters GLUT and IR expression which worsen hyperglycemia status. The synthesis of placental hormones is altered by hyperglycemia. Deficient synthesis of PRL and calcitriol (the active form of vitamin D) has been reported in GDM placentae, whereas hPL synthesis is increased. These placental hormonal changes, and probably estrogens and progestins, compromise pancreatic gene expression of GLUT2, glucokinase, insulin, survivin, cyclin D2, and Bcl6; all these genes are related to b-cells proliferation and survival. Because of reduced b-cells expansion, hyperglycemia worsens. Additionally, deficient synthesis of calcitriol abates placental expression of antimicrobial peptides related to innate defense, increasing mother and fetus vulnerability to infections in the perinatal period. Another endocrine dysregulation is derived from IGF-I overactivation in the placenta, which explains increased placental growth and glucose uptake. The establishment of a chronic inflammatory milieu in the placenta results in: (i) Increased production of pro-inflammatory cytokines, chemokines, and adipokines; (ii) Increased placental infiltration of M1 macrophages, cytotoxic neutrophils, and T cells; (iii) Deregulated production of pro-angiogenic and anti-angiogenic factors; and (iv) Increased lipid peroxidation and synthesis of ROS precursors. Even if angiogenesis is induced, vessels in the placenta are thickened and immature. Overall, altered vessels formation, hyperglycemia, and oxidative stress induce endothelial dysfunction. Furthermore, the serum inflammatory profile is evidenced by high levels of pro-inflammatory cytokines, chemokines, and adipokines as well as a higher presence of cytotoxic monocytes, neutrophils, and T cells. Hyperglycemia and over-activation of IGF-I signaling results in increased fetal growth which may contribute to macrosomia. Finally, hyperglycemia and oxidative stress produce pancreatic overstress in the fetus, causing an increased risk for T2DM development later in life. Experimental and clinical evidence indicates that GDM fetuses present a marked neural pro-oxidative environment which may lead to neural, motor and behavior disturbances. AMPs: antimicrobial peptides; GDM: Gestational diabetes mellitus; GLUT: Glucose transporter; hPL: human placental lactogen; IGF-I: Insulin-like Growth Factor 1; IR: Insulin receptor; N: Neutrophils; PRL: Prolactin; ROS: Reactive oxygen species; T: T lymphocytes; T2DM: type 2 Diabetes mellitus; TG: triglycerides; VD: vitamin D.

**Table 1 ijms-22-08087-t001:** Main reactive oxygen and nitrogen species altered in GDM.

Metabolite	Description	GDM Status	Reference
Protein carbonyls	Created in response to oxidative stress and are produced in the process of protein carbonylation.	Increased in plasma at 24–28 WOG	[284,285]
Nitrotyrosine	Superoxide O_2_^−^• is produced by the electroreduction of molecular oxygen, is highly reactive, with a short half-life. It is metabolized to hydrogen peroxide by the high activity of superoxide dismutase. In highly vascularized tissues such as the human placenta, superoxide can combine with nitric oxide to form the very reactive and damaging ROS, peroxynitrite (ONOO^−^•). ONOO^−^• reacts with hydroxyl groups on serine, tyrosine, and threonine residues to form nitrotyrosine adducts.	Increased in the blood (24–28 WOG)	[284,286]
Advanced glycation end products (AGEs)	AGEs are formed by a combination of glycation and oxidation reaction. AGEs are strongly associated with diabetic complications of pregnancy that have increased levels of glucose as well as ROS and are associated with complications of diabetes such as retinopathy, nephropathy, and neuropathy. AGEs are formed due to the non-enzymatic glycation of proteins, lipids, and nucleic acids during hyperglycemia. It has the potential to damage vasculature by modifying the substrate or utilizing AGEs and Receptor of AGE (RAGE) interaction. AGEs such as Ne-carboxymethyl lysine (CML) are produced under oxidative conditions.	AGEs are elevated in maternal serum. CML is elevated in plasma (24–30 WOG)	[287,288,289,290]
Oxidized low-density lipoproteins (Ox-LDL)	The formation of ox-LDL involves the oxidation of both protein and lipid components.	Elevated in maternal plasma	[291,292]
4 -hydroxynonenal (4-HNE)	A product of lipid peroxidation that can modify proteins and have effects on tissue survival.	Elevated in placental and uterine tissues	[283,293]
Malondialdehyde (MDA)	MDA is a marker of lipid peroxidation and oxidative stress and is formed as the result of lipid peroxidation of polyunsaturated fatty acids.	Elevated maternal plasma levels at 28 WOG	[292]
Oxidized DNA	DNA is vulnerable to oxidation and undergoes a continuous cycle of damage and repair in living cells. The nucleoside oxidation marker 8-oxo-7,8 dihydro-2 deoxyguanosine (8OH-dG) is used as a biomarker of oxidative stress excreted in urine.	8OH-dG level in the urine was 26% higher in women who subsequently developed GDM	[294,295]
Glutathione (GSH) and Glutathione peroxidase (GPX)	Under healthy conditions, cells have up to 98% of total GSH in their reduced form. GSH acts as an electron donor to reduce unstable ROS in a process that permits the formation of oxidized GSH. An altered ratio reduced GSH/oxidized GSH is an indicator of oxidative stress. While reduced GSH can directly prevent non-enzymatic oxidation of thiol groups; it also acts as a co-substrate for the antioxidant enzyme GPX. GPX can remove active peroxides, but this results in the conversion of reduced GSH to oxidized GSH.	Reduced GSH in blood at term. Diminution of GPX in blood	[296,297,298]

WOG: weeks of gestation.

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
