# Peer review of "Immunoendocrine Dysregulation during Gestational Diabetes Mellitus: The Central Role of the Placenta"

_ijms, 2021, doi:10.3390/ijms22158087_

Round 1
Reviewer 1 Report
The manuscript “Immunoendocrine dysregulation during gestational diabetes mellitus: the central role of the placenta” from Andrea Olmos-Ortiz et al. is a review covering extensive field related to endocrine, immune and metabolic functions of placenta. The paper fails to identify novelty and does not postulate objectives to produce such an extensive review.
Major comments
- Despite the fact that authors postulate GDM subtypes regarding the severity of glucose metabolism derangement and subsequent event. need for pharmacotherapy, the rest of the text do not differentiate and presents GDM as a homogenous phenotype. In fact, with nowadays standard of screening and care, there are few perinatal pathologies and complications. The long-term impact for mother and offspring in terms of metabolic risks (incl. foetal programming) is another aspect.
- Subheadings are definitely helpful, but still, the text is dense. Authors combine clinical observations with molecular findings which makes difficult to follow the message.
- Conclusions should be much better elaborated in pathophysiological and clinical terms and gaps for future research identified.
Author Response
Reviewer 1
- The manuscript "Immunoendocrine dysregulation during gestational diabetes mellitus: the central role of the placenta" from Andrea Olmos-Ortiz et al. is a review covering extensive field related to endocrine, immune and metabolic functions of placenta. The paper fails to identify novelty and does not postulate objectives to produce such an extensive review.
Response: We agree with the comments of the reviewer regarding the importance of highlight the novelty of this review. Therefore, we would like to mention that this review aims to examine and understand the placenta's role as a vital organ that acts as the interface between maternal and fetal metabolisms impacted by a pathological condition as GDM.
This approach is, in fact, the novelty of the review. As you know, there is a vast corpus of evidence showing the systemic impact of the GDM. However, our objective was to concentrate efforts on the placenta as a target organ directly impacted by a hyperglycemic fetal-maternal environment.
In immunologic and endocrine terms, this transient tissue must support, and at the same time partially compensate, the changes associated with meta-inflammation, innate defense, hormone production, oxidative stress, and angiogenesis.
To answer this suggestion, we have integrated the scope/aim in a new paragraph at the end of the introduction section.
Major comments
- "Despite the fact that authors postulate GDM subtypes regarding the severity of glucose metabolism derangement and subsequent events. Need for pharmacotherapy, the rest of the text does not differentiate and presents GDM as a homogenous phenotype".
Response: We appreciate the reviewer's suggestion regarding the necessity of homologating the concepts and subtypes of GDM. In this sense, considering that this review focuses on the placenta as an immune-endocrine organ impacted by the hyperglycemic environment, we decided, to gain clarity, to maintain a homogeneous phenotype of GDM along with the manuscript.
- "In fact, with nowadays standard of screening and care, there are few perinatal pathologies and complications".
Response: We agree with the reviewer. Therefore, the section on outcomes was entirely reviewed and rewrite.
- "The long-term impact for mother and offspring in terms of metabolic risks (incl. foetal programming) is another aspect"
Response: Foetal programming is undoubtedly an essential aspect during a condition as GDM. The impact of a pathologic uterine environment can induce multiples epigenetics changes that might affect the fetus's life in the short, medium, and long term.
However, from our point of view, approaching this aspect of the GDM requires a systematic review explicitly focused on this aspect. However, the outcomes section is now briefly mentioned as an important area of interest and study.
5. "Subheadings are definitely helpful, but still, the text is dense".
Response: We agree with the reviewer; therefore, a complete revision of the manuscript, with particular attention to the grammatical structure, had been done. Additionally, the manuscript has been reviewed for two native English speakers to improve the document understandably.
6. "Authors combine clinical observations with molecular findings which makes difficult to follow the message".
Response: We thank the reviewer for pointing this out. We also want to add that the integration of both clinical and molecular approximations was an original objective in this review. However, as we mentioned previously, we complete the manuscript's revision to improve the lecture of the ideas and concepts.
- "Conclusions should be much better elaborated in pathophysiological and clinical terms, and gaps for future research identified.
Response: We agree with the reviewer. Therefore, we rewrote the conclusions section:
“The functional interplay between the placenta and the maternal adipose tissue, glycemic control, and overall maternal metabolism, requires a highly controlled equilibrium that, if disturbed, may lead to GDM. Known risk factors for GDM include overweight/obesity, lack of physical activity, prediabetes, and genetic predisposition. In this pathology, maternal hyperglycemia, carbohydrates intolerance, dysfunction of beta-pancreatic and endothelial cells, as well as insulin resistance, disturb placental structure and functions. This disturbance favors a metainflammatory environment associated with increased production of inflammatory cytokines, adipokines, and oxidative reactive species, leading to an abnormal endocrine, immune and antioxidant phenotype. This maternal/placental immunoendocrine dysregulation affects the mother as well as the fetus's health in the short and long term. The severity of complications relates to an earlier onset of GDM and correlates inversely with the degree of glycemic control. Therefore, it is of utmost importance to understand the pathophysiology of GDM to develop intervention and prevention strategies, including the orientation of pregnant women to eat healthier foods and exercising. In Figure 2, we propose a multi-organ scheme integrating the role of the placenta in the immunoendocrine environment dysregulated during GDM.”
Reviewer 2 Report
This review is informative and provides great depth of knowledge related to adverse mechanisms affecting placenta in GDM. However, what is missing is the link between these aberrant mechanisms and the subsequent effects on the placental functionality and pregnancy complications such as preeclampsia, preterm birth, delivery complications (shoulder dystocia) etc. The metabolic and inflammatory signalling as a result of GDM is well explained and comprehensively discussed however the important aspects of GDM and its effects on the placental vasculature/integrity leading to complications and preterm birth are missing. This link between aberrant mechanisms and subsequent effects is missing and the authors should address this gap. One of the largest studies on the planet, the HAPO study and its multiple papers, describe pregnancy complications extensively and this is a good start to link those aberrant mechanisms with subsequent pregnancy complications (https://care.diabetesjournals.org/content/35/4/780.long).
There are other concerns related to the paper in its present form:
At the start GDM is described as "failure to balance insulin secretion" when insulin resistance is a key component.
The sentence describing "chronic metabolic stress over the pancreas during pregnancy..." is somewhat confusing. How is pregnancy chronic?
When treatments of GDM were discussed, care should be taken as some international guidelines use metformin as a first line treatment. A meta-analysis from 2018 shows metformin to be more effective at preventing preeclampsia (leading cause of death in pregnancy) in GDM whereas insulin appear to increase the risk of preeclampsia in GDM through inducing weight gain. This shoud be discussed. https://onlinelibrary.wiley.com/doi/abs/10.1111/dme.13523
TNF-α was discussed a lot as well as the STAT pathways but there was no mention of the NFkB pathway?
Page 9, lines 390-393 discussed increase in inflammation and anti-angiogenic factors in GDM placentae - care should taken with this as diabetic placentae are often found to be hypervascularised and leaky with increase in angiogenic factors and decrese in anti-angiogenic factors that often leads to endothelial dysfunction as described in this recent paper: https://www.frontiersin.org/articles/10.3389/fendo.2021.650328/full#B72
In fact, in GDM (and other pre-gestational diabetes) hypoxia and proangiogenic phenotype is often observed in placentae leading to impaired integrity
https://pubmed.ncbi.nlm.nih.gov/23573311/ https://pubmed.ncbi.nlm.nih.gov/11043866/
Section 3.2 seems out of place and should be moved to Section 4.
Why was only vitamin D discussed in section 4.2? Surely there are other therapeutic options with anti-inflammatory effects. Unless there is a strong rational for only discussing vitamin D, this appears as incomplete or biased account of potential treatments for GDM-induced placental inflammation and immune response.
Section 6 needs to be much better crafter. It would be better to link aberrant mechanisms to different complications associated with GDM throughout the paper and list complication at in the introduction.
Figure 1&2 are really informative. Table is also well designed.
Other minor points:
Subtitles should be more informative and crafted better
There are a number of very long sentences that are difficult to follow so make them more concise.
What is the difference between pre-natal and peri-natal - in one sentence there is a mention of "prenatal and perinatal periods..."
There are some references missing e.g. when pre-gestational cause of GDM are described or when complications of GDM are listed etc.
The quality of english in the manuscript should be imporved
Placentas should be referred to as "placentae"
"Codified"?
"pathways awaken"?
"subexpression" - should be downregulation or decrease in experession
"weightier than control"?
"longitudinal growth along with life" - do you mean with age?
"which" is use a lot throughout the review
Author Response
Response to Reviewer 2.
Comments and Suggestions for Authors
- This review is informative and provides great depth of knowledge related to adverse mechanisms affecting placenta in GDM. However, what is missing is the link between these aberrant mechanisms and the subsequent effects on the placental functionality and pregnancy complications such as preeclampsia, preterm birth, delivery complications (shoulder dystocia) etc. The metabolic and inflammatory signalling as a result of GDM is well explained and comprehensively discussed however the important aspects of GDM and its effects on the placental vasculature/integrity leading to complications and preterm birth are missing. This link between aberrant mechanisms and subsequent effects is missing and the authors should address this gap. One of the largest studies on the planet, the HAPO study and its multiple papers, describe pregnancy complications extensively and this is a good start to link those aberrant mechanisms with subsequent pregnancy complications (https://care.diabetesjournals.org/content/35/4/780.long). (10.2337/dc11-1790)
Response: We thank the reviewer for pointing this out. We now understand the necessity of a link between GDM and the different fetal and maternal outcomes. Therefore, we have reviewed and rewritten this section using all references suggested.
- There are other concerns related to the paper in its present form: At the start GDM is described as "failure to balance insulin secretion" when insulin resistance is a key component.
Response: The reviewer is correct; the failure in pancreatic insulin secretion is a secondary complication derived firstly from insulin resistance. Therefore, the text was modified as follows: " It is well accepted that a key event in the onset of GDM is the maternal peripheral insulin resistance " (Page 1).
- The sentence describing "chronic metabolic stress over the pancreas during pregnancy..." is somewhat confusing. How is pregnancy chronic?
Response: We apologize; it was inadequate to choose the word "chronic" in the limited period of pregnancy. The intention was to describe this time of hyperglycemia / metabolic stress as a sustained period along gestation. For this reason, we decided to replace the word chronic for sustained / persistent along manuscript.
- When treatments of GDM were discussed, care should be taken as some international guidelines use metformin as a first line treatment. A meta-analysis from 2018 shows metformin to be more effective at preventing preeclampsia (leading cause of death in pregnancy) in GDM whereas insulin appear to increase the risk of preeclampsia in GDM through inducing weight gain. This shoud be discussed. https://onlinelibrary.wiley.com/doi/abs/10.1111/dme.13523
Response: We agree with the reviewer. Therefore, to resolve this suggestion, we restructured the paragraph about the pharmacological treatment of GDM, including the additional references, which the reviewer suggested, and others more. This change was highlighted in the tracked version of the manuscript.
The complete paragraph read as follows:
“However, other Societies including ACOG, German Diabetes Association, German Society of Gynecology and Obstetrics, and The Society for Maternal-Fetal Medicine recommend metformin instead [1,12]. Recent studies showed a lower risk for preeclampsia, macrosomia, neonatal hypoglycemia and hypertensive disorders as well as better outcomes in maternal weight gain and glycemic control. No difference was observed in rates of caesarean section, neonatal respiratory distress and preterm birth compared to insulin treatment alone [13–17]. There is insufficient evidence on the long-term effects of prenatal exposure to metformin (especially because it crosses the placenta). Two recent studies showed no difference in growth and development in children of metformin-treated and insulin-treated mothers over a four-year period [18,19]. More long-term studies are needed to understand the long-term effects of metformin during pregnancy.”
- TNF-α was discussed a lot as well as the STAT pathways but there was no mention of the NFkB pathway?
Response: This suggestion is correct, now we include a breve description of these pathways will improve the manuscript.
We included the following two paragraphs about NFkB signaling in the manuscript (Page 10):
“Undoubtedly, the signaling of NFkB is the main regulator of inflammatory pathways in normal and GDM placentae. After TNF-a binding with their receptor TNFR1, the adaptor protein TRADD is recruited and associated with the death domain of TNFR1. TRADD acts like platform binding for TRAF2 and RIP adaptor proteins which eventually activated the TAK1 kinase to phosphorylate and activate the IKK complex formed by the catalytic subunit IKKa and IKKβ, and the regulatory subunit NEMO.
The IKK complex phosphorylates the IkB proteins that are constitutively bound to NFkB, keeping this factor in the cytosol. The serine phosphorylation of IkB proteins promotes their ubiquitination and proteolytic degradation by the proteasome, free
allowing the nuclear translocation of NFkB [140,141]. The NFkB upregulates target genes that encoded pro-inflammatory cytokines, inducing a chronic inflammatory loop that contributes to the development of insulin resistance.
NFkB can be activated by endogenous molecules released during tissue damage and oxidative stress, including debris from apoptotic, saturated fatty acids, heat shock proteins, advanced glycation products (AGEs) which are recognized by the TLR-4 receptor [142]. In GDM and maternal hyperglycemia there is a positive association with an increase of TLR-4 and NFkB signaling in the placenta [143,144]. TLR-4-induction of NFkB signaling in the placenta is an important mechanism that is altered during gestational diabetes; however, further studies are needed to elucidate the involvement of innate immunity in trophoblast functionality.”
- Page 9, lines 390-393 discussed increase in inflammation and anti-angiogenic factors in GDM placentae - care should taken with this as diabetic placentae are often found to be hypervascularised and leaky with increase in angiogenic factors and decrese in anti-angiogenic factors that often leads to endothelial dysfunction as described in this recent paper: https://www.frontiersin.org/articles/10.3389/fendo.2021.650328/full#B72In fact, in GDM (and other pre-gestational diabetes) hypoxia and pro-angiogenic phenotype is often observed in placentae leading to impaired integrity
https://pubmed.ncbi.nlm.nih.gov/23573311/ https://pubmed.ncbi.nlm.nih.gov/11043866/
Response: Concerning this observation, we would like to comment that a completely new section entitled “5. Endovascular changes in GDM: Endothelial dysfunction and overstimulation of placental angiogenesis” was included in this new version. Our intention is to offer the readers an integrated view of the main findings in this field.
- Section 3.2 seems out of place and should be moved to Section 4.
Response: We consider that this suggestion is pertinent, and we made this change in the new version of the manuscript.
8. Why was only vitamin D discussed in section 4.2? Surely there are other therapeutic options with anti-inflammatory effects. Unless there is a strong rational for only discussing vitamin D, this appears as incomplete or biased account of potential treatments for GDM-induced placental inflammation and immune response.
Response: Thank you for pointing this out. It was not our intention to portrait vitamin D as a therapeutic option for GDM. Instead, we wanted to highlight the critical participation of calcitriol, the vitamin D hormonal form, in regulating the placental immune milieu and insulin resistance.
In particular, considering the dual effect of calcitriol in immune regulation (potent anti-inflammatory and stimulator of antimicrobial peptides), we decided to place this section immediately after discussing the alterations in the adaptive and innate immunity that occur in the GDM placentas.
For clearness, we separated this section from the previous one and changed the subtitle for one that better reflects its content.
- Section 6 needs to be much better crafter. It would be better to link aberrant mechanisms to different complications associated with GDM throughout the paper and list complication at in the introduction.
Response: Thank you very much for this recommendation. We evaluated the possibility of placing the complications associated with GDM in other sections throughout the paper. However, considering the importance of these complications, we feel that a whole "stand-alone" section describing the topic of adverse perinatal outcomes in patients with GDM is the best way to conclude our paper. Nevertheless, we have made an effort to craft this section better to make it clearer to the reader.
10. Figure 1&2 are really informative. Table is also well designed.
Response: We appreciate reviewer found data resumed in Figures and tables as useful.
Other minor points:
- Subtitles should be more informative and crafted better
Response: We agree; therefore, a complete revision has been made.
There are a number of very long sentences that are difficult to follow so make them more concise.
Response: After revision of the English style, the document is now more concise.
- What is the difference between prenatal and perinatal - in one sentence there is a mention of "prenatal and perinatal periods..."
Response: We appreciate this comment. The prenatal period is generally considered as the time before birth (from conception until delivery), whereas the perinatal period includes the period immediately before and after birth (from the 22nd week of gestation and ends 1 week after birth, although some authors extend from the 20th week of gestation and ends 4 weeks after birth) (https://meteor.aihw.gov.au/content/index.phtml/itemId/327314).
In consideration of both definitions, we consider that the concept perinatal fits better with our idea. Therefore, the text was modified as follows: " Although it is a transient, GDM effects can last beyond the perinatal period and impact the health of mother and fetus in both short- and long-term [2–4]. (Page 1)
4. There are some references missing e.g. when pre-gestational cause of GDM are described or when complications of GDM are listed etc.
Response: The references were revised to correct all mistakes and omissions.
- The quality of English in the manuscript should be improved.
Response: We agree with the reviewer; therefore, a complete revision of the manuscript, with particular attention to the grammatical structure, had been done. Additionally, the manuscript has been reviewed for two native English speakers to improve the document understandably.
- Placentas should be referred to as "placentae"
Response: The manuscript was carefully revised, and the concept "placentae" was applied.
- "Codified"?,"pathways awaken"?
Response: We apologize for these mistakes. All of them have been corrected.
- "subexpression" - should be downregulation or decrease in expression
Response: This mistake has been corrected.
- "weightier than control"?
Response: The phrase was changed to "Reported placental abnormalities in GDM patients include: increased placental weight". (Page 2)
10. "longitudinal growth along with life" - do you mean with age?
Response: This phrase was corrected as follows: " It is well known that the growth hormone (GH)-IGF-I axis is the major regulator of longitudinal growth along life” (Page 7).
- "which" is use a lot throughout the review
Response: After the complete revision of the manuscript, this issue was corrected.
Round 2
Reviewer 1 Report
All reviewer's comments have been addressed adequately. The paper should be accepted in current